# Sacrum morphology supports taxonomic heterogeneity of "*Australopithecus africanus*" at Sterkfontein Member 4

Cinzia Fornai [1,2✉], Viktoria A. Krenn[1,2], Philipp Mitteroecker[3], Nicole M. Webb [1,4] & Martin Haeusler [1]

The presence of multiple *Australopithecus* species at Sterkfontein Member 4, South Africa (2.07–2.61 Ma), is highly contentious, and quantitative assessments of craniodental and postcranial variability remain inconclusive. Using geometric morphometrics, we compared the sacrum of the small-bodied, presumed female subadult *Australopithecus africanus* skeleton Sts 14 to the large, alleged male adult StW 431 against a geographically diverse sample of modern humans, and two species of *Pan, Gorilla*, and *Pongo*. The probabilities of sampling morphologies as distinct as Sts 14 and StW 431 from a single species ranged from 1.3 to 2.5% for the human sample, and from 0.0 to 4.5% for the great apes, depending on the species and the analysis. Sexual dimorphism and developmental or geologic age could not adequately explain the differences between StW 431 and Sts 14, suggesting that they are unlikely to be conspecific. This supports earlier claims of taxonomic heterogeneity at Sterkfontein Member 4.

[1] Institute of Evolutionary Medicine, University of Zurich, Zurich, Switzerland. [2] Department of Evolutionary Anthropology, University of Vienna, Vienna, Austria. [3] Department of Evolutionary Biology, University of Vienna, Vienna, Austria. [4] Department of Palaeoanthropology, Senckenberg Research Institute and Natural History Museum Frankfurt, Frankfurt, Germany. ✉email: cinzia.fornai@univie.ac.at

Traditionally, all early hominin fossils from Taung, Sterkfontein Member 4 and Makapansgat have been attributed to *Australopithecus africanus*, although several authors noted the remarkably high morphological variability within these assemblages[1–11]. This led Clarke[1–3,12] to propose the presence of a second *Australopithecus* species at Sterkfontein Member 4, which he attributed to *A. prometheus*, a species name originally given to fossils from Makapansgat[13]. Recent studies of the StW 573 "Little foot" skeleton from Sterkfontein Member 2 renewed debates on the functional biology and taxonomy of the Plio-Pleistocene hominins from South Africa[12,14–16]. Nevertheless, the presence of two closely related *Australopithecus* taxa at Sterkfontein Member 4 is not widely accepted[17,18] because of conflicting interpretations and the fragmentary preservation of the fossils. Moreover, quantitative studies focused mainly on the craniodental remains and rarely considered morphological variation within a broad comparative setting.

Sts 14[19–22] and StW 431[22–27] are the best-preserved, albeit incomplete skeletons from Sterkfontein Member 4. In Sts 14, the unfused apophyses of the iliac crests and ischial tuberosities and the partially fused epiphyseal plates of the sacral alae suggest an age of about 15 years based on modern human standards[24,28,29]. Its acetabulum, which already reached adult dimensions, predicts a body size of 25.4 kg[30], and the pubic morphology suggests a female sex[20,24]. StW 431, an adult individual that probably weighed considerably more than 40 kg[30,31], was possibly of male sex based on its greater robusticity and larger body size. Both Sts 14 and StW 431 are conventionally attributed to *A. africanus*. However, the StW 431 sacrum is narrower and more elongated while Sts 14 is comparatively gracile and wide relative to its small sacral body[27,32] (Fig. 1). The upper lateral angles of the transverse processes are prominent in both StW 431 and Sts 14. However, they project superiorly in StW 431 while in Sts 14 they are laterally oriented, resulting in a smooth and elongated superior aspect of the sacral alae[24]. This contrasts to the sacrum of A.L. 288-1 (*Australopithecus afarensis*), which lacks well-developed upper lateral angles of the transverse processes[33].

Here, we quantify the morphological differences of the sacrum between Sts 14 and StW 431 using a comparative approach. We explore how the magnitude of their shape differences compares to within-species variation of modern humans and extant great ape species while taking into account factors such as sexual dimorphism and differences in individual and geologic age. Accordingly, we use geometric morphometrics based on 113 3D landmarks (Supplementary Data 1) representing the preserved sacral vertebrae of StW 431 and Sts 14 to investigate a geographically diverse sample of juvenile and adult modern humans ($n = 74$) and an extensive sample of extant great apes ($n = 94$, from six different species). As Sts 14 and StW 431 do not patently show sacral segmentation anomalies, we confine our comparative sample to specimens without lumbosacral transitional vertebrae. We apply two different morphometric approaches, one in which we register the landmark configurations by a Generalized Procrustes Analysis (GPA) of all 113 landmarks, and one in which all

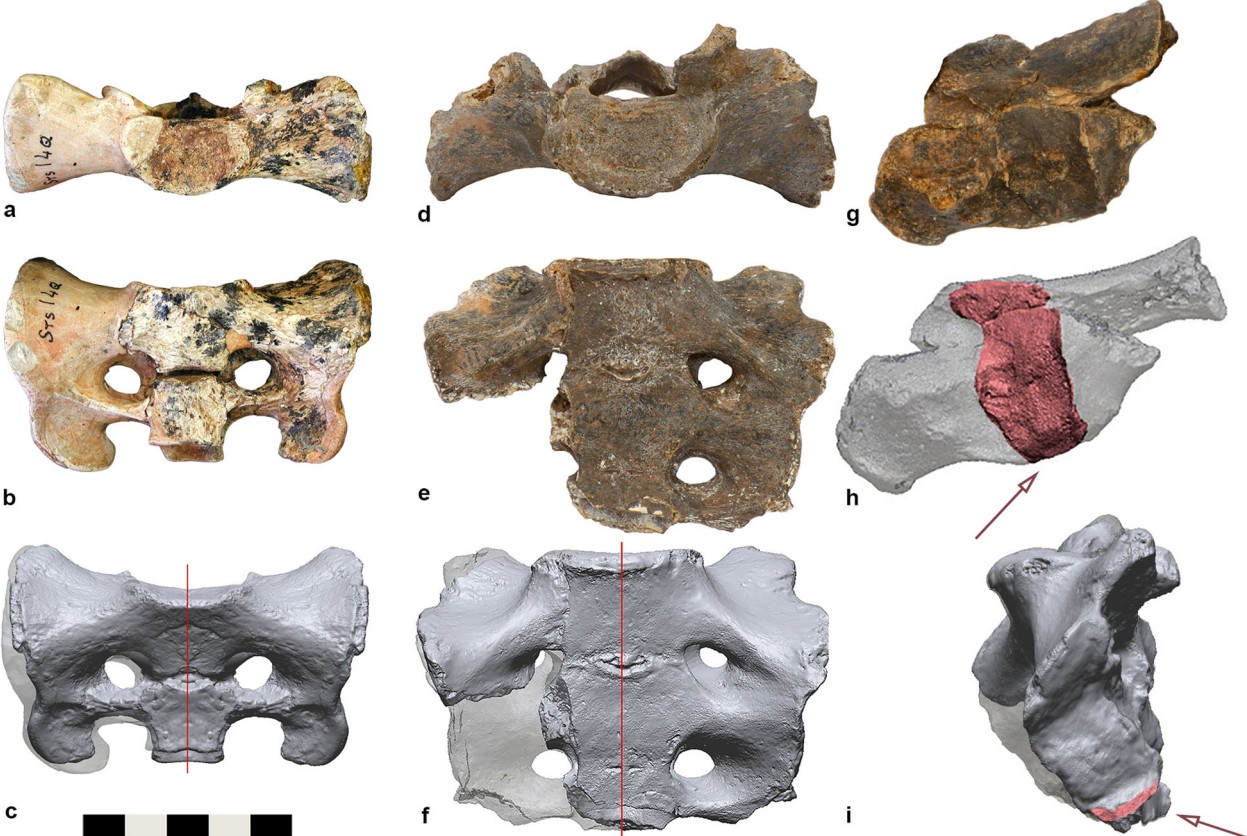

**Fig. 1 The sacrum of Sts 14 and StW 431, both traditionally attributed to *Australopithecus africanus*. a** Photograph of the Sts 14 sacrum, superior view and **b** anterior view. **c** 3D-surface model of the reconstructed Sts 14 sacrum produced by mirroring the left side with respect to the mid-sagittal plane (red line), thereby removing Robinson's reconstruction in plaster of Paris[21] (shown in transparent). **d** Photograph of the StW 431 sacrum, superior view and **e** anterior view. **f** Reconstructed 3D-surface model of StW 431 (in transparent) obtained by mirroring the left side of the sacrum at the mid-sagittal plane (red line). **g** Photograph and **h** 3D-surface model of the left ilium fragment of StW 431; the sacroiliac joint surface is coloured in red. **i** The most inferior portion of the sacral auricular surface (arrow) was restored using the auricular surface of the ilium.

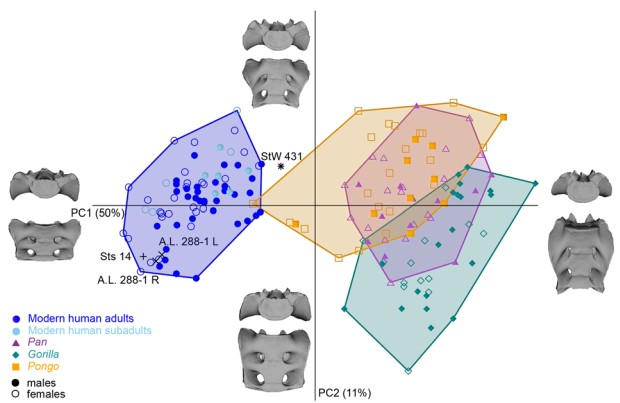

**Fig. 2 PCA plot of the shape coordinates of the first two sacral vertebrae and auricular surface after a Generalized Procrustes Analysis of the complete sample.** Sts 14 and StW 431 (both attributed to *Australopithecus africanus*) plot at opposite sides of the modern human distribution (blue circles = adults; sky-blue circles = subadults), while the two reconstructions of the A.L. 288-1 (*A. afarensis*) sacrum are close to Sts 14. The great apes (medium-orchid triangles = *Pan*; dark-cyan diamonds = *Gorilla*; orange squares = *Pongo*) are separated from the hominins along PC1. PC1 is driven by the overall height-to-width ratio of the sacrum, and PC2 represents the orientation and relative antero-posterior width of the sacral alae. Closed symbols = males, open symbols = females.

landmarks are registered based on a GPA of the 13 landmarks on the body of the first sacral vertebra only (Procrustes fit on a subset of landmarks[34,35]). This second geometric morphometric analysis is chosen because the relative size and shape of the sacral alae vary greatly with respect to the sacral body. The datasets originating from the GPA are analysed both in shape space and form space. To assess the morphological heterogeneity between the StW 431 and Sts 14 sacra, their Procrustes distance is compared to all pairwise differences within our extant species sample. Our investigation reveals that the morphological differences between Sts 14 and StW 431 could not be fully explained by sexual dimorphism or differing developmental or geologic age, suggesting that taxonomic heterogeneity may contribute to the morphological variability within Sterkfontein Member 4 *Australopithecus*.

## Results and discussion

In the principal component analysis (PCA) of the shape coordinates based on the GPA of all landmarks, Sts 14 and StW 431 plotted at the opposite sides of the modern human distribution, whereas all extant great apes were clearly separated from modern humans along PC1 (accounting for 50% of the total variance) (Fig. 2, Supplementary Data 2, and Supplementary Movie 1). StW 431 was reminiscent of great apes in having a narrower and more elongated sacrum compared to that of humans, whereas Sts 14 possessed relatively wider and superiorly flatter alae. Nevertheless, StW 431 was overall more similar to modern humans than great apes (see the quadratic discriminant analysis in Supplementary Note 1). The two reconstructions of the A.L. 288-1 sacrum plotted close to Sts 14. Shape variation along PC2 (11%) was driven by the supero-inferior orientation of the sacral alae relative to the sacral body. StW 431 differed along PC2 from Sts 14 and A. L. 288-1 for its caudally directed linea terminalis. PC3 (6%) reflected changes in the antero-posterior orientation of the sacral alae and did not add notably to the differences between StW 431 and Sts 14. Some *Pongo* specimens were intermediate between the human and great ape clusters, while one *Pongo pygmaeus* female overlapped with the *Homo* cluster because of its relatively broad shape. *Pongo* and *Gorilla* separated from one another quite well,

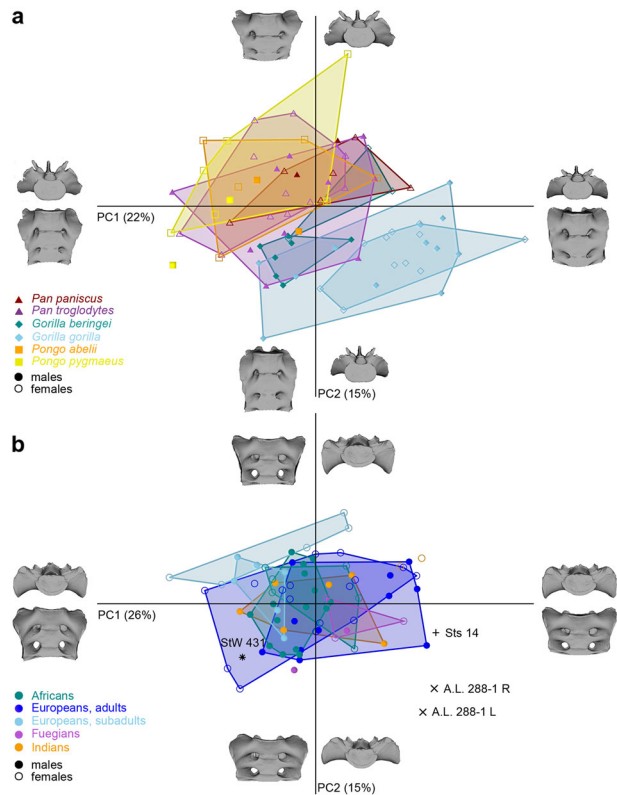

**Fig. 3 PCA plot of the Procrustes shape coordinates of the first two sacral vertebrae and auricular surface after a Generalized Procrustes Analysis of great apes and modern humans. a** PCA plot of great apes, labelled by species and sex (vermillion triangles = *Pan paniscus*; medium-orchid triangles = *P. troglodytes*; dark-cyan diamonds = *Gorilla beringei*; sky-blue diamonds = *G. gorilla*; orange squares = *Pongo abelii*; yellow squares = *P. pygmaeus*). Within each genus, the species and sexes largely overlap, except for *G. beringei*. **b** PCA plot for the upper portion of the sacrum (GPA) in modern humans and *Australopithecus*, labelled by sex, age, and ethnicity (dark-cyan circles = Africans; blue circles = European adults; sky-blue circles = European subadults; medium-orchid circles = Fuegians; orange circles = Indians). Along PC1, the sacral portion of the linea terminalis varies from more horizontal, as in Sts 14 and A.L. 288-1, to more caudally oriented, as in StW 431. A.L. 288-1 differs from StW 431 and Sts 14 along PC2 for its posterior orientation of the alae. The various modern human populations do not separate, but the subadult females tend to differ from the adults along PC2, while StW 431 and Sts 14 differ along PC1. Closed symbols = males, open symbols = females.

whereas *Pan* overlapped extensively with the two other ape genera.

The different species within the genera *Pan*, *Gorilla*, and *Pongo* were undistinguishable based on sacral shape (Fig. 3a), and so were the different modern human populations (Fig. 3b). The sacral shape of the subadult modern humans—with ages comparable to that of Sts 14—tended to separate from adult modern humans along PC2, while Sts 14 and StW 431 diverged mainly along PC1. Even though modern human males and females overlapped in the PCA plot, their mean shapes differed significantly in shape space ($p = 0.0005$). Size-related shape variation accounted for 12.3% of total sacral shape variation within the full sample but for only 3.3% within the hominin sample (see also Supplementary Fig. 1).

Since sacrum morphology of *Australopithecus* more closely resembled that of *Homo*, we repeated the geometric morphometric analysis after excluding the great apes (Fig. 3b). In this

**Table 1 Percentages of the pairwise Procrustes distances that exceeded the Procrustes distance between StW 431 and Sts 14.**

| Groups | GPA, shape space | | GPA, form space | | Procrustes fit on S1 body, shape space | |
|---|---|---|---|---|---|---|
| | m/f pairs | all pairs | m/f pairs | all pairs | m/f pairs | all pairs |
| Modern human adults ($n = 63$) | 1.5 | 1.3 | 3.8 | 3.7 | 2.6 | 2.1 |
| Modern human adults and subadults ($n = 74$) | 2.5 | 2.4 | 3.4 | 3.3 | 2.6 | 2.3 |
| *Pan* ($n = 30$) | 24.1 | 21.8 | 6.0 | 6.9 | 0.9 | 1.1 |
| *Gorilla* ($n = 33$) | 18.8 | 18.8 | 54.2 | 39.4 | 5.4 | 4.2 |
| *Pongo* ($n = 31$) | 26.7 | 28.8 | 39.5 | 28.2 | 2.4 | 4.5 |
| *Pan troglodytes* ($n = 22$) | 24.2 | 22.1 | 0.8 | 0.4 | 0.0 | 0.0 |
| *Gorilla gorilla* ($n = 23$) | 15.2 | 15.8 | 47.7 | 32.0 | 4.5 | 3.6 |
| *Pongo pygmaeus* ($n = 24$) | 30.5 | 31.2 | 43.8 | 31.5 | 3.9 | 6.2 |

The computations were performed for male–female pairs only as well as for all pairwise comparisons.
*m/f* male–female, *S1* first sacral vertebra.

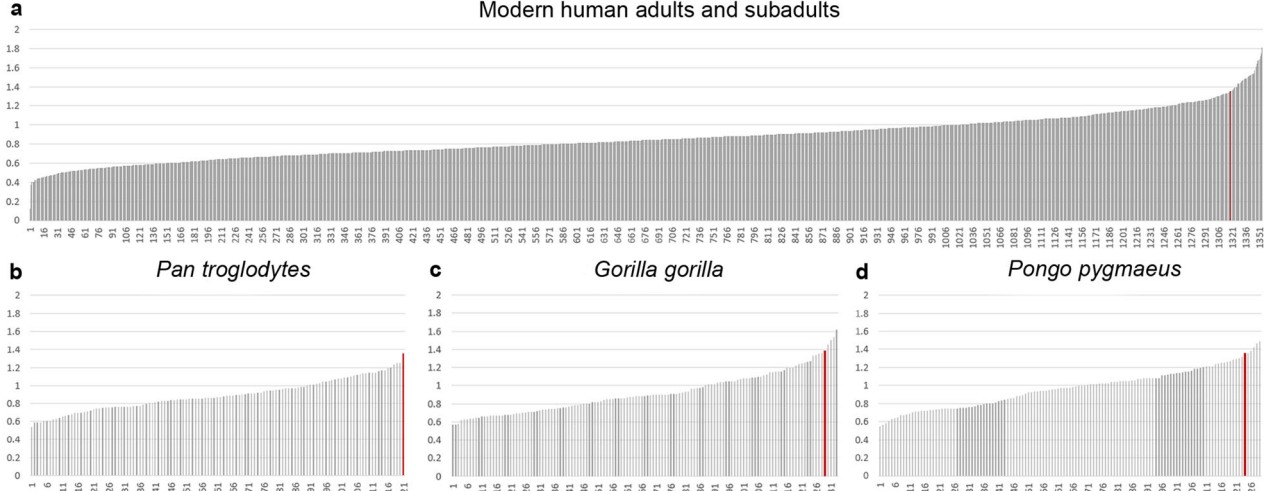

**Fig. 4 Column distribution of the male–female pairwise Procrustes distances after Procrustes fit based on 13 landmarks on the first sacral vertebra.** The red lines indicate the Procrustes distance between Sts 14 and StW 431. **a** In modern humans only 35 out of 1353 Procrustes distances (2.6%) exceeded that between Sts 14 and StW 431. **b** In *Pan troglodytes* 0% (0 out of 120) distances exceeded that between the fossils, **c** in *Gorilla gorilla* 4.5% (6 out of 132), and **d** in *Pongo pygmaeus* 3.9% (5 out of 128).

reduced sample, the main axis of variation was driven by the supero-inferior orientation of the alae, which again distinguished StW 431 from Sts 14. The A.L. 288-1 sacrum deviated from modern humans and the other *Australopithecus* specimens for its more posteriorly oriented alae and the weakly developed upper lateral angles of the transverse processes (PC2). The Procrustes distance, a measure of overall shape difference, between StW 431 and Sts 14 was among the largest observed distances within the modern human sample, including both adults and subadults (Table 1, Fig. 4, and Supplementary Data 2). Only 2.5% of the Procrustes shape distances between all pairwise male–female comparisons within modern humans were higher than the Procrustes distance between Sts 14 and StW 431. A very similar result (2.4%) was obtained when all pairwise Procrustes distances of modern humans, regardless of their sex, were considered. Procrustes distances in form space (comprising differences in both shape and size; see Supplementary Fig. 2) resulted in slightly higher percentages (3.4% and 3.3%, respectively). Within great apes, a greater percentage of pairwise Procrustes distances exceeded that between Sts 14 and StW 431 (Table 1).

To explore the contribution of sexual dimorphism to sacral shape, we performed the analysis only with modern human adults, and Sts 14 and StW 431 (Supplementary Fig. 3, Supplementary Data 2, and Supplementary Movie 2). Sexual dimorphism was represented by PC2 and PC3, despite a large overlap between males and females, whereas they showed similar range of variation along PC1. Importantly, the differences between Sts 14 and StW 431 along PC2 and PC3 resembled the average sexual dimorphism in humans, both in pattern and magnitude. However, the two fossils mainly differed along PC1, which is unrelated to sexual dimorphism. Along this PC, 1.0% of all pairwise differences and 1.1% of male–female pairwise differences exceeded the distance between Sts 14 and StW 431.

To better characterize the relative dimensions of the sacral alae, we compared the sacral shape of StW 431 and Sts 14 to those of the recent species after the registration of the landmark configurations based on the corpus landmarks only. The resulting PCA plot (Supplementary Fig. 4) was similar to that presented above in Fig. 2 for the separation of hominins from great apes along PC1, but showed a more pronounced overlap between the great apes. The percentages of pairwise Procrustes distances in *Homo* exceeding the distance between Sts 14 and StW 431 ranged from 2.1 to 2.6%. Importantly, the corresponding percentages in the great ape species were lower than for the GPA of all landmarks,

ranging from 0% in *P. troglodytes* to 4.5% in *G. gorilla* for male–female pairs (Table 1).

**Sexual dimorphism, allometry, and individual age**. In modern humans, sexual dimorphism of the sacrum is low compared to that of other parts of the pelvis[36]. Nonetheless, we found a statistically significant shape dimorphism in the human sacrum, which is also reflected in the shape differences between StW 431 and Sts 14 and might even be a more general primate pattern[37]. However, the overall shape difference between the two fossils remained very high even when compared to the morphological distances of all possible male–female pairwise combinations. As expected, we did not observe considerable sexual dimorphism in great ape sacrum shape. Size-related shape differences within hominins were of small magnitude (3.3% allometric shape variation) and, hence, are unlikely to account for the differences in sacral shape between StW 431 and Sts 14. It is also unlikely that the shape differences between the fossils result from differences in individual age because in our PCA analysis (Fig. 3) the vector between StW 431 and Sts 14 (which closely aligns with PC1) was almost perpendicular to the vector of average human ontogenetic shape change (close to PC2). Moreover, if Sts 14 had developed to full adult age, the lateral epiphyseal plates would have fused. This would have resulted in an even wider sacrum, thereby increasing the observed morphological distance from StW 431.

**Geological age**. According to the most recent U-Pb dating of flowstones[38,39] the maximum period for the accumulation of the Sterkfontein M4 is between 2.61 and 2.07 Ma. Thus, Sts 14 and StW 431 could theoretically differ up to 540,000 years in chronological age. The exact provenience of Sts 14 is unknown but there are claims that it originated from sediments close to the top of Member 4. This skeleton was said to be found within a single block not far from Sts 5 ('Mrs. Ples')[40,41] which in turn might have come from the vicinity of the flowstone topping Sterkfontein Member 4[38]. However, flowstones are often post-depositional infillings of voids within the cave sediments and thus only provide a minimum age for the fossils[42]. On the other hand, almost all fragments of the StW 431 skeleton were found in two adjacent square yards at a mean depth of 7 m below datum, while many other Member 4 fossils were recovered from deeper deposits. Parts of the StW 431 skeleton were vertically distributed between a depth of 6.5 m below datum (most of the vertebrae and right arm) and a depth of 9.0 m (right pubis fragment)[24,43]. This distribution reflects a complex taphonomy of the StW 431 skeleton, which might be explained by disturbance of a talus cone by later collapses[44,45]. Thus, although the dating of the StW 431 remains is challenging, there is no indication that they are markedly different in age from Sts 14.

In any case, a possible difference in geological age between StW 431 and Sts 14 can only partially, if at all, account for the differences in sacrum shape. The human populations in our sample completely overlapped in the PCA of sacrum shape, even though some of them have an evolutionary depth of about 260–350 ka[46]. Moreover, *P. pygmaeus* and *P. abelii* diverged about 400 ka years ago[47] but were indistinguishable in the PCA. Even *Pan troglodytes* and *P. paniscus* were comparable in sacrum shape although they diverged 1.6 Ma ago, with extensive gene flow until at least 200 ka ago[48]. *Gorilla gorilla* and *G. beringei* diverged 1.75 Ma ago, with substantial gene flow until at least 20 ka ago[49], but also largely overlapped in sacrum shape. Thus, sacrum shape appears to be evolutionarily conserved, presumably due to stabilizing selection imposed by biomechanical and obstetric demands[50].

**Conclusions**. Since Sts 14 and StW 431 preserve only the first two and a half sacral vertebrae, it is impossible to completely rule out transitional vertebrae in either of these specimens, hence this aspect should be investigated further. In fact, Sts 14 shows a segmentation anomaly at the thoracolumbar transition[21,51], such border shifts are frequent in hominin fossils and are often associated with border shifts at the lumbosacral junction[52].

In conclusion, our results show that, under the assumption that the *A. africanus* sacrum was as variable in shape as that of modern humans and extant great ape species, it is unlikely though possible that Sts 14 and StW 431 belong to a single species, thereby supporting earlier claims of taxonomic heterogeneity at Sterkfontein Member 4. Yet, as neither Sts 14 nor StW 431 is associated with craniodental remains, it remains impossible to infer which one of these partial skeletons, if any, belongs to *A. africanus*.

## Methods

**The fossil and reference samples**. We compared the partial sacrum of StW 431 and Sts 14 to a sample of 157 sacra from adult modern humans and great apes of known sex, as well as the complete sacrum of the presumed female A.L. 288-1 (*A. afarensis*). Our geographically diverse modern human sample comprised 63 individuals including Central Europeans ($n = 28$), Western Africans ($n = 13$), Khoe-Sān ($n = 7$) and Pygmies ($n = 2$), Indians ($n = 8$), and Fuegians ($n = 5$). The great ape sample included *P. paniscus* ($n = 8$), *P. troglodytes* ($n = 22$), *G. beringei* ($n = 10$), *G. gorilla* ($n = 23$), *P. abelii* ($n = 7$), and *P. pygmaeus* ($n = 24$) (Table 2). Since Sts 14 died prior to fusion of the lateral epiphyseal plate and of the ring apophysis of the superior surface of the sacral body[24,53,54], we also included six male and five female juvenile modern humans (Central Europeans, $n = 10$, Khoe-Sān, $n = 1$) with a developmental age similar to that of Sts 14 (16–17 years). Specimens with damaged or eroded surface or with patent asymmetry were excluded, as well as individuals with lumbosacral transitional vertebrae. 3D-surface models of the sacra were obtained using a high-resolution optical 3D-surface scanner (QTSculptor PT-M4, www.polymetric.de)[55]. In instances where CT data were available, the meshes were generated by segmentation using the software Amira (www.fei.com). The skeletal collections and the source of the surface models are listed in Table 2.

**Virtual reconstruction of the *Australopithecus* specimens**. The StW 431 sacrum preserves most of the first two and a half sacral vertebrae on the left side, while the right side is more damaged. The most inferior portion of the left auricular surface is

**Table 2 List of fossil sacra and modern comparative material.**

|  | Individuals/taxa | Females | Males | Collections |
|---|---|---|---|---|
| *Australopithecus* | Sts 14q | 1 |  | a |
|  | StW 431h |  | 1 | b |
|  | A.L. 288-1an | 1 |  | c |
| Modern humans | Adults | 28 | 35 | d, e, f, g, h, i, j |
|  | Subadults | 5 | 6 | k |
| Great apes | *Pan paniscus* | 6 | 2 | l |
|  | *Pan troglodytes* | 12 | 10 | d, h, m, n, o, p, q |
|  | *Gorilla beringei* | 2 | 8 | d, h, m, o |
|  | *Gorilla gorilla* | 11 | 12 | d, h, m, n, o |
|  | *Pongo abelii* | 5 | 2 | d, h |
|  | *Pongo pygmaeus* | 16 | 8 | d, h, l, n, q |
| Total modern comparative sample ($N = 168$) |  | 85 | 83 |  |

a: Ditsong National Museum of Natural History, Pretoria; b: Evolutionary Studies Institute, University of the Witwatersrand, Johannesburg, South Africa; c: National Museum of Ethiopia, Addis Ababa, Ethiopia; d: Anthropological Institute and Museum, University of Zurich, Switzerland; e: Department of Anthropology, Natural History Museum Vienna, Austria; f: Department of Evolutionary Anthropology, University of Vienna, Austria; g: Institute of Evolutionary Medicine, University of Zurich, Switzerland; h: Laboratory of Prehistoric Archaeology and Anthropology, University of Geneva, Switzerland; i: Museum of Natural History, University of Florence, Italy; j: Smithsonian National Museum of Natural History, Washington, USA; k: Hospital Timone, Marseille, France; l: Royal Museum for Central Africa, Tervuren, Belgium; m: Zoological Museum, University of Zurich, Switzerland; n: Digital Morphology Museum, KUPRI, Kyoto University, Japan; o: Department of Zoology, Natural History Museum Vienna, Austria; p: Museum of Primatology, University of California, San Diego; q: Natural History Museum Basel, Switzerland.

also broken off and was restored based on the shape of the auricular surface of the ilium (Fig. 1). This resulted in an auricular surface ~2 mm longer than the one originally preserved. Afterwards, the restored left side of the fossil was mirrored with respect to the mid-sagittal plane to replace the incomplete right side. The Sts 14 sacrum preserves the left side of the first two sacral vertebrae. We reconstructed it virtually after removing the right side previously restored with plaster of Paris by Robinson[21] by mirroring the preserved left side with respect to the mid-sagittal plane. Since the A.L. 288-1 sacrum is taphonomically distorted, we obtained two symmetrised versions of the fossil, one for the mirrored right side and the other for the mirrored left side.

**Statistics and reproducibility.** The landmark configuration was conceived to represent the preserved aspects of the StW 431 and the Sts 14 sacrum. It was therefore confined to the first two sacral vertebrae (which are developmentally homologous among all taxa considered in this study)[51] and consisted of 29 anatomical landmarks, 36 semilandmarks on five curves, and 48 surface semilandmarks (Supplementary Fig. 5). The curves described the margins of the superior sacral surface and of the auricular surfaces, as well as the superior and posterior aspects of the alae. The anterior and superior aspects of the upper portion of the sacrum were represented by surface semilandmarks. The posterior surface is usually highly variable, and is heavily reconstructed in Sts 14. Therefore, only few landmarks and curve semilandmarks were gathered on the dorsal side of the sacrum. The geometric morphometric analysis was repeated both with and without the surface semilandmarks to explore the contribution of the interlandmark surface patches. Since the anterior surface of the sacrum between the chosen landmarks is rather smooth, we did not obtain relevant differences in the outcomes and thus present only the results for the complete landmark configuration.

Standard geometric morphometric analyses were performed using principal component analysis (PCA) of the Procrustes shape coordinates both in shape space and in form space, the latter by augmenting the shape coordinates with the natural logarithm of centroid size (lnCS)[34,56–58]. The landmarks were collected in Viewbox 4 (www.dhal.com), and analysed in Evan Toolbox (www.evan-society.org). The sliding of the semilandmarks was based on the minimum bending energy criterion after relaxation against the consensus shape. The geometric morphometric analysis after Procrustes fit based on a subset of 13 landmarks on the body of the first sacral vertebra was performed in the R software environment[59] using a code written explicitly for this purpose. The software PAST was used for generating the PCA plots.

The morphological differences between StW 431 and Sts 14 were evaluated with respect to the rest of the sample by running a pairwise comparison of the Procrustes distances (i.e. the square root of the summed squared differences between the corresponding shape coordinates of two landmark configurations) after the GPA analysis in both shape and form space, and after a Procrustes fit based on the subset of 13 landmarks on the body of the first sacral vertebra. The percentages of Procrustes distances higher than that observed between StW 431 and Sts 14 were computed for all pairwise comparisons and separately also for male–female pairs only, both on the genus and species level, if at least 20 individuals could be included. An intra- and interobserver error assessment was performed by C.F. and V.A.K. confirming high precision of the landmark configurations (the highest Procrustes distance between all repeats was lower than 2700 out of the 2701 pairwise Procrustes distances between the 74 measured human sacra). Variation in sacral size was assessed by box plots of lnCS in the R software environment[59] (Supplementary Fig. 2). The influence of size on shape (i.e. allometry) was evaluated by regressing the Procrustes shape variables on lnCS. The male and female group mean differences were tested for the modern humans using a permutation test of the Procrustes distances (10,000 random permutations). Since StW 431 plotted between modern humans and *Pongo* in the PCA analysis after GPA for the entire sample, the likelihood of classifying StW 431 to either group was evaluated using Quadratic Discriminant Analysis (Supplementary Note 1).

**Reporting summary.** Further information on experimental design is available in the Nature Research Reporting Summary linked to this paper.

## Data availability
The authors declare that the data supporting the findings of this study are available within the paper and its supplementary information files.

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

## Acknowledgements

We are grateful to Ronald J. Clarke for discussing the Southern African australopithecine morphology and taphonomy, and for providing comments on an earlier version of the paper. For the fossil material we thank Stephany Potze, Ditsong National Museum of Natural History; Bernhard Zipfel, Evolutionary Studies Institute, University of the Witwatersrand; Metasebia Endalamaw and Yared Assefa, National Museum of Ethiopia. For the modern human and great ape samples we are indebted to: Jocelyne Desideri, Laboratory of Prehistoric Archaeology and Anthropology, University of Geneva; Harald Wilfing, Katrin Schaeffer, and Katarina Matiasek, Department of Evolutionary Anthropology, University of Vienna; Marcia Ponce de Leon, Anthropological Institute and Museum, University of Zurich; Darrin Lunde, Smithsonian Institution, National Museum of Natural History; Eduard Winter, Sabine Eggers, Karin Wilschke-Schrotta, Frank Zachos, Nicole Grunstra, Natural History Museum Vienna; Monica Zavattaro, Jacopo Moggi-Cecchi, Natural History Museum, University of Florence; Emmanuel Gilissen, Royal Museum for Central Africa; Barbara Oberholzer, Zoological Museum, University of Zurich; Louise Corron, Department of Anthropology, University of Nevada; Loïc Costeur, Natural History Museum Basel; Digital Morphology Museum, KUPRI, Kyoto University; Museum of Primatology CARTA, University of California, San Diego. We thank Fred L. Bookstein for discussion of the statistics and for helping writing the codes for the Quadratic Discriminant Analysis and for the Procrustes fit on a landmark subset. Alexander Gruber and Matthias Diem helped coding utilities on Spyder, Python. This project was financially supported by the Swiss National Science Foundation (grant Nos 31003A_156299/1 and 31003A_176319).

## Author contributions

M.H. initiated and organized the project. C.F. and M.H. designed the research protocol. C.F., V.A.K. and M.H. gathered the image data. C.F. and V.A.K. segmented the CT data, generated the surface models, and collected the landmark data. C.F. virtually reconstructed the fossil specimens. C.F. and P.M. performed the statistical analyses supported by N.M.W. All authors discussed the results. C.F. and M.H. wrote the manuscript; all authors edited it. C.F. compiled tables and figures.

## Competing interests

The authors declare no competing interests.
