## [Peer Review File · Communications Biology]

Reviewers' comments:

Reviewer #1 (Remarks to the Author):

The taxonomic heterogeneity of Sterkfontein Member 4 remains an important question in human evolution. Fornai and colleagues have conducted a 3DGM analysis of the sacrum in order to evaluate whether the morphological differences between two supposed *Au. africanus* fossils (Sts 14 and StW 431) are large enough to merit their attribution to different species. This is a technically sound and well written contribution to the ongoing debate. I have relatively minor suggestions for improvement of the manuscript:

Abstract:

The authors have succinctly summarized the results in the discussion without overstating them. It would be nice to incorporate that phrasing into the abstract as well. E.g., re-wording the last sentence to say something along the lines of:

"These findings suggest that it is possible, but unlikely, that Sts 14 and StW 431 belong to a single species, thereby supporting earlier claims of taxonomic heterogeneity at Sterkfontein Member 4."

Main text:

P10-11: The authors state "The human populations in our sample completely overlapped in the PCA of sacrum shape, even though they diverged at least 260 to 350 ka years ago 42."

- I would be cautious to discuss the 'divergence of human populations' and their genetic separation as this can open a controversial can of worms (particularly in this context). Remove this sentence and stick to comparisons with the pan-pan and gorilla-gorilla divergences instead.

P14: "Since StW 431 plotted between modern humans and Pongo in the PCA analysis after full GPA for the entire sample, the likelihood of classifying StW 431 to either groups was evaluated using Quadratic Discriminant Analysis."

- Typo, should be changed to: "classifying StW 431 to either group"

SOM:

P1: "StW 431 could be morphologically allied with either Homo and Pongo."

- Typo, should be changed to "Homo or Pongo"

P1: "using a script prepared and supplied by Fred L. Bookstein."

- For this to be a repeatable study, could this script also be included as a supplemental file to the publication?

P1: "The $p \sim 0.05$ threshold for the hypothesis that StW 431 belongs to Homo is 6.83, and that for the hypothesis that StW 431 belongs to Pongo is 0.146. Intermediate values reflect equivocal results. The likelihood ratio (LR) for the hypothesis that StW 431 belongs to Homo is 33.90 for PC1-PC2 and 38.44 for PC2-PC3."

- While the rest of the paper was well written, this particular passage was difficult to follow. In a simplified explanation, what is the conclusion we should draw from this analysis? More detail would be useful.

Reviewer #2 (Remarks to the Author):

Thank you for inviting me to revise this manuscript. This is an interesting study by Fornai and colleagues assessing the morphology of three australopithecine sacra by means of geometric morphometric techniques. I enjoyed and I learned a lot from the several readings of this manuscript. I have mainly minor comments and a couple of major suggestions, specially related to the methods, although what is a minor and a major comment lastly depends on the authors' criteria. In any case, I hope my comments and suggestions are helpful to improve the (already good) manuscript.

"Minor" comments

- Although the "problem" of the morphological variability in *Australopithecus* is well defined in the first paragraph of the introduction, I personally think that the aim of this study is not clearly stated. In the last paragraph of the Introduction, the authors indicate that they did a GMM study of the sacra of Sts 14 and StW 431 but they don't indicate what the purpose is, if they have any hypothesis or what they expect to find.

- After 3D surface scanning and CT scanning, did the authors fuse and post process the scans to obtain the final 3D models? In my opinion, the article lacks information about how the data was collected and post processed (e.g. details of the CT scanning, post post-processing such as smoothing, mesh decimation, etc.) although I don't know what the standards of this journal are in terms of detailed methods (maybe in Supplementary Methods?).

- As the sample comes from archaeological collections, based on my experience I would say that it is likely that some of the bones had missing parts (especially those of Khoe San, Pygmies and Fuegians). Were all the sacra complete or did the authors estimate missing landmarks in Viewbox? If the latter, please, explain. Did the authors use the symmetric or the asymmetric component of the sacral morphology? As they used a human specimen to design a template that they used to measure also great apes, did they relax the configurations against a mean? Please, see pages 83-85 in Gunz, P., Mitteroecker, P., & Bookstein, F. L. (2005). Semilandmarks in three dimensions. In *Modern morphometrics in physical anthropology* (pp. 73-98). Springer, Boston, MA.

- The authors perform a great part of the analyses in open source software, and that is brilliant. Indeed, the analyses they did in EVAN toolkit can be performed in R too (wink!). They indicate that some analyses are performed in "R software package" (e.g. page 13, line 269). I think they mean the "R environment", because there are lots of R packages with different names. Could the authors please indicate the names of the packages (and version and citation where necessary) they used to do the analyses in R?

- Relatedly, the R community benefits (I am sure that the authors too) from the altruism of people who share their codes in open source repositories. Given that this is an open access journal and they shared the landmark data, I encourage them to share the R code they used. There are plenty of repositories (GitHub, Open Science Framework) and even the supplementary section of the journal, where the authors can share their R code. Of course I am not saying I won't recommend acceptance of the article if the code is not shared, but this kind of practices promotes transparency and reproducibility in science.

- I see that the authors used different approaches in their analyses. The first approach is related to the area/region of the sacrum. As I understand from Supplementary Figure 5, on the one hand they analyze 113 landmarks and semilandmarks, and on the other hand they analyze 13 landmarks of the body of the first sacral vertebra (according to page 5, lines 70-71). However, I can't see what those 13 landmarks represent in the sacrum morphology. Could the authors please remark the subset of 13 landmarks in Supplementary Figure 5? The second approach that I see is (according to page 13, lines 260-264), on the one hand, the analysis of the whole landmark configuration with surface

semilandmarks, and on the other hand, the analysis without the surface semilandmarks. I am a huge advocate of this kind of comparisons in geometric morphometrics but if I am correct and I am not missing anything, the authors did not report the results of the second approach; they only say that “did not obtain relevant differences in the outcomes and thus presented the results for the complete landmark configuration”. I personally think it would be convenient to report overall (not necessarily detailed) results of the two analyses (with surface sml vs without surface sml) in supplementary information as Mitteroecker and Bookstein (2008) did in their section “Alternative analyses” of Mitteroecker, P., & Bookstein, F. (2008). The evolutionary role of modularity and integration in the hominoid cranium. *Evolution: International Journal of Organic Evolution*, 62(4), 943-958.

- The authors measured only the first two sacral vertebrae but, if I am correct, they did not explain why. Maybe this is obvious because Sts 14 and StW 431 only preserve that region, but this is not indicated in the manuscript. Relatedly, there is no mention to the different number of sacral vertebrae present in the different species analyzed. If the approach of analyzing the first two sacral vertebrae avoids the problem of lack of homology between the sacra of these taxa, the authors should indicate it.

- What do the authors mean by “full GPA”? A GPA with rotation, translation and scaling? A GPA where the centroid size is reduced to $\cos(p)$ (see Zelditch et al. 2012, Chapter 4)? A GPA superimposing the full configuration (113 landmarks and semilandmarks)? By the way, what do you mean by “full space” on page 13, line 274?

- The authors briefly mention PC3 (e.g. page 7, line 136, page 8 line 138) but if I am correct, this plot is not in the manuscript.

- In page 10, lines 183-186 the authors mention perpendicular “vectors” in the PCA analysis but I don’t see any vector in any of the PCA plots. Are these vectors a way to connect means within the PCA? Are statistical vectors?

- Page 10, line 185. “In our PCA”. Which PCA? Please, indicate Figure.

- Page 11, line 225. “Our worldwide sample 63 modern humans...”. Is there a word missing?

- Page 14, line 285. “using Quadratic Discriminant Analysis”. Please, add a reference to Supplementary Note” after this.

“Major” comments

- What is the role of the auricular surface in this study? I can see that the morphology is quantified through the landmark configuration (113) but its variation is overlooked in the manuscript. It is surprising as Stw 431 is characterized by its robusticity and body size so assessing this surface would be of interest, basically because the authors reconstructed this part as shown in Figure 1. If the authors consider that this is not relevant for the study, please, explain why (in the text).

- Did the authors assess the measurement error? Intra- and/or inter-?

- I wonder why it of interest to include Pongo and Gorilla (especially Pongo) in this study. I suspect it is way to standardize the analyses including all great apes, but I wonder why the inclusion of our closest living relatives of the genus *Pan* is not enough for this kind of comparative studies. Gorilla could be considered by some authors as a scaled version of *Pan* (similar locomotor mode, etc.) but Pongo is a completely different thing and its inclusion needs to be justified. The bulk of the analyses of work are PCA and PCA is very sensitive to the specimens included and their variation so a PCA with only *Homo* and *Pan* would look different from a PCA including all great apes.

To increase the transparency and openness of the reviewing process promoted by the journal, below I sign my report to authors. Lastly, my apologies if any of my comments does not make sense, are redundant or if I missed something after the several readings. I am looking forward to reading a revised version of the manuscript.

Nicole Torres-Tamayo

Reviewer #3 (Remarks to the Author):

This paper employed a 3D landmark-based geometric morphometric approach to quantify sacral morphology in extant large hominoids (human, in which to place the differences in sacral morphology observed between Sts 14 and StW 431 to test the hypothesis that these specimens do not represent conspecific individuals. The authors analysis shows that the morphology of these specimens do not exceed the range of variation seen within a single modern specie, albeit they would fall at extreme ends of the range of variation. Therefore, the data do not falsify the hypothesis that these specimens represent a single species. Furthermore, given relatively small sample sizes (Our worldwide sample 63 modern humans including Central Europeans (N=28), Western Africans (N=13), Khoe-Sān (N=7) and Pygmies (N=2), Indians (N=8), and Fuegians (N=5). The great ape sample included *G. beringei* (N=10), *G. gorilla* (N=23), *P. paniscus* (N=8), *P. troglodytes* (N=22), *P. abelii* (N=7), and *P. pygmaeus* (N=24). The differendes between these specimens represents typical morphology associated with some segmental variation within species. As the data and analysis do not falsify the hypothesis of conspecificity, they do not support the authors conclusions. I therefore recommend this paper be rejected.

Dear Reviewers,

Thank you for the enthusiastic yet meticulous review of our work.

Changes are highlighted in red both here and in the manuscript. Deletions in the manuscript are stricken out.

Yours faithfully,

Cinzia Fornai for all authors

	Reviewer #1 (Remarks to the Author)	Reply to Reviewer #1
1.1	The taxonomic heterogeneity of Sterkfontein Member 4 remains an important question in human evolution. Fornai and colleagues have conducted a 3DGM analysis of the sacrum in order to evaluate whether the morphological differences between two supposed Au. africanus fossils (Sts 14 and StW 431) are large enough to merit their attribution to different species. This is a technically sound and well written contribution to the ongoing debate. I have relatively minor suggestions for improvement of the manuscript: Abstract: The authors have succinctly summarized the results in the discussion without overstating them. It would be nice to incorporate that phrasing into the abstract as well. E.g., re-wording the last sentence to say something along the lines of: “These findings suggest that it is possible, but unlikely, that Sts 14 and StW 431 belong to a single species, thereby supporting earlier claims of taxonomic heterogeneity at Sterkfontein Member 4.”	We agree with the reviewer that our conclusions should have been made clearer in the abstract and because of word limitations we now write (l. 22-25): ‘Differences in developmental or geologic age or sexual dimorphism could not explain the differences between StW 431 and Sts 14, suggesting that they are unlikely conspecific. This supports earlier claims of taxonomic heterogeneity at Sterkfontein Member 4.’
1.2	Main text: P10-11: The authors state “The human populations in our sample completely overlapped in the PCA of sacrum shape, even though they diverged at least 260 to 350 ka years ago 42.” • I would be cautious to discuss the ‘divergence of human populations’ and their genetic separation as this can open a controversial can of worms (particularly in this context). Remove this sentence and stick to comparisons with the pan-pan and gorilla-gorilla divergences instead.	We acknowledge that the phrasing we used (divergence of human populations) might raise controversy although that was not our intention. Nevertheless, we believe that the evolutionary depth of the morphological diversity within modern humans is still an important aspect for the comparison between StW 431 and Sts 14. As such, we have reformulated this sentence as follows (l. 210-211): “The human populations in our sample completely overlapped in the PCA of sacrum shape, even though some of them have an

		evolutionary depth of about 260 to 350 ka ⁴⁵ .” In addition, we changed “similarly” at the beginning of the next sentence to “moreover” to differentiate it better from speciation events in great apes.
1.3	P14: “Since StW 431 plotted between modern humans and Pongo in the PCA analysis after full GPA for the entire sample, the likelihood of classifying StW 431 to either groups was evaluated using Quadratic Discriminant Analysis.” • Typo, should be changed to: “classifying StW 431 to either group”	Corrected (l. 311)
1.4	SOM: P1: “StW 431 could be morphologically allied with either Homo and Pongo.” • Typo, should be changed to “Homo or Pongo”	Corrected (SOM I. 17-18)
1.5	P1: “using a script prepared and supplied by Fred L. Bookstein.” • For this to be a repeatable study, could this script also be included as a supplemental file to the publication?	The code will be available upon request to the corresponding author and with permission of Professor Fred Bookstein. We rephased to (l. 291-292): ‘...was performed in the R software environment ⁵⁸ using a code written for this purpose.’ Credit to Professor Bookstein is given in the Acknowledgments.
1.6	P1: “The p ~0.05 threshold for the hypothesis that StW 431 belongs to Homo is 6.83, and that for the hypothesis that StW 431 belongs to Pongo is 0.146. Intermediate values reflect equivocal results. The likelihood ratio (LR) for the hypothesis that StW 431 belongs to Homo is 33.90 for PC1-PC2 and 38.44 for PC2-PC3.” • While the rest of the paper was well written, this particular passage was difficult to follow. In a simplified explanation, what is the conclusion we should draw from this analysis? More detail would be useful.	It now sounds (SOM I. 22-24): ‘The likelihood ratio (LR) for the hypothesis that StW 431 belongs to Homo is 33.90 for PC1-PC2 and 38.44 for PC2-PC3, thereby indicating clear morphological affinity to Homo ’.
	Reviewer #2 (Remarks to the Author)	Reply to Reviewer #2
2.1	Thank you for inviting me to revise this manuscript. This is an interesting study by Fornai and colleagues assessing the morphology of three australopithecine sacra by means of geometric morphometric techniques. I enjoyed and I learned a lot from the several readings of this manuscript. I have mainly minor comments and a couple of major suggestions, specially related to the methods, although what is a	We added (l. 64-70): ‘Here, we quantify the morphological differences of the sacrum between Sts 14 and StW 431 in a comparative setting to explore whether the magnitude of their shape differences is compatible with within-species variation of modern humans and extant great ape species when factors such as sexual variation, ontogenetic development, possible differences geologic age and,

	minor and a major comment lastly depends on the authors' criteria. In any case, I hope my comments and suggestions are helpful to improve the (already good) manuscript. "Minor" comments - Although the "problem" of the morphological variability in Australopithecus is well defined in the first paragraph of the introduction, I personally think that the aim of this study is not clearly stated. In the last paragraph of the Introduction, the authors indicate that they did a GMM study of the sacra of Sts 14 and StW 431 but they don't indicate what the purpose is, if they have any hypothesis or what they expect to find.	eventually, taxonomic heterogeneity are taken into account. Accordingly, we use geometric morphometrics based on 113 3D landmarks representing the preserved sacral vertebrae of StW 431 and Sts 14 ...' and below (l. 74-83): 'We apply two different morphometric approaches, one in which we register the landmark configurations by a Generalized Procrustes Analysis (GPA) of all 113 landmarks, and one in which all landmarks are registered based on a GPA of the 13 landmarks on the body of the first sacral vertebra only (Procrustes fit on a subset of landmarks ^{34,35}). This second geometric morphometric analysis is chosen because the relative size and shape of the sacral wings vary greatly with respect to the sacral body. The datasets originating from the GPA are analysed both in shape space and form space. To assess the morphological differences between the StW 431 and Sts 14 sacra their Procrustes distance are compared to all pairwise differences within our extant species sample.'
2.2	- After 3D surface scanning and CT scanning, did the authors fuse and post process the scans to obtain the final 3D models? In my opinion, the article lacks information about how the data was collected and post processed (e.g. details of the CT scanning, post post-processing such as smoothing, mesh decimation, etc.) although I don't know what the standards of this journal are in terms of detailed methods (maybe in Supplementary Methods?).	The scanning device we used, and its associated software, are able to produce high resolution triangular meshes comprised of around 500,000 to 1,000,000 polygons. Moreover, the final surface model represents the entire object without holes needing to be filled during the postprocessing of the surface, as is often the case with other scanner models. We didn't report these technical aspects in the manuscript since we feel that these additional notations would not add relevant information to the Methods. It also would not enhance or inhibit reproducibility with its inclusion so it is superfluous in this case.
2.3	- As the sample comes from archaeological collections, based on my experience I would say that it is likely that some of the bones had missing parts (especially those of Khoe San, Pygmies and Fuegians). Were all the sacra complete or did the authors estimate missing landmarks in Viewbox? If the latter, please, explain. Did the authors use the symmetric or the asymmetric component of the sacral morphology? As they used a human specimen to design a template that they used to measure also great apes, did they relax the configurations against a mean? Please, see pages 83-85 in Gunz, P., Mitteroecker, P., &	We specify now that (l. 248-249): "Specimens with damaged or eroded surface or with patent asymmetry were excluded, as well as individuals with lumbosacral transitional vertebrae." Moreover, we placed only few landmarks in the more critical, and surely morphologically highly variable, dorsal region. As we did not separate symmetric from asymmetric variation, the sliding of the semilandmarks was performed against a general consensus. The relevant text in Methods now reads (l. 287-289): 'The landmarks were collected in Viewbox 4

	Bookstein, F. L. (2005). Semilandmarks in three dimensions. In Modern morphometrics in physical anthropology (pp. 73-98). Springer, Boston, MA.	(www.dhal.com), and analysed in the Evan Toolbox (www.evan-society.org). The sliding of the semilandmarks was based on the minimum bending energy criterion after relaxation against the consensus shape.'
2.4	- The authors perform a great part of the analyses in open source software, and that is brilliant. Indeed, the analyses they did in EVAN toolkit can be performed in R too (wink!). They indicate that some analyses are performed in "R software package" (e.g. page 13, line 269). I think they mean the "R environment", because there are lots of R packages with different names. Could the authors please indicate the names of the packages (and version and citation where necessary) they used to do the analyses in R?	Please note that the open-source code for the EVAN Toolbox is freely available from their website https://www.evan-society.org/. We used the EVAN Toolbox for the traditional GM analysis (which, indeed, can be performed in R, too). The visualization tools are a forte of the EVAN toolbox, which we therefore preferred over available R packages for GM analysis. However, the GM analysis based on the GPA on a subset of landmarks cannot be performed in the EVAN toolbox. We instead used a customized code written with the help of Fred L. Bookstein specifically for this analysis. This point has been clarified in the manuscript, too (l. 287-292): 'The landmarks were collected in Viewbox 4 (www.dhal.com) and analysed in the Evan Toolbox (www.evan-society.org). The sliding of the semilandmarks is based on the minimum bending energy criterion after relaxation against the general consensus. The geometric morphometric analysis after partial Procrustes fit based on a subset of 13 landmarks on the body of the first sacral vertebra was performed in the R software environment⁵⁸ using a code written for this purpose. The PAST software was used for generating the PCA plots.' We replaced 'R software package' with 'R software environment' throughout the manuscript (l. 291, 306, SOM 18).
2.5	- Relatedly, the R community benefits (I am sure that the authors too) from the altruism of people who share their codes in open source repositories. Given that this is an open access journal and they shared the landmark data, I encourage them to share the R code they used. There are plenty of repositories (GitHub, Open Science Framework) and even the supplementary section of the journal, where the authors can share their R code. Of course I am not saying I won't recommend acceptance of the article if the code is not shared, but this kind of practices promotes transparency and reproducibility in science.	The code will be available upon request to the corresponding author and with permission of Professor Fred Bookstein.

2.6	- I see that the authors used different approaches in their analyses. The first approach is related to the area/region of the sacrum. As I understand from Supplementary Figure 5, on the one hand they analyze 113 landmarks and semilandmarks, and on the other hand they analyze 13 landmarks of the body of the first sacral vertebra (according to page 5, lines 70-71). However, I can't see what those 13 landmarks represent in the sacrum morphology. Could the authors please remark the subset of 13 landmarks in Supplementary Figure 5? The second approach that I see is (according to page 13, lines 260-264), on the one hand, the analysis of the whole landmark configuration with surface semilandmarks, and on the other hand, the analysis without the surface semilandmarks.	Both GM analyses were performed using 113 landmarks. The analyses differ in the way the landmark configurations are registered. One is a classical GPA, where all landmarks are used to superimpose (register) and scale the specimens. The second approach superimposes and scales the specimens using only a subset of landmarks (in our case 13 landmarks on the first sacral body), while the remaining 100 landmarks follow passively. The 13 landmarks used for the superimposition are now shown in Supplementary Figure 5 c) and d). We have rephrased the relevant paragraph in the manuscript as such (l. 74-83): "We apply two different morphometric approaches, one in which we register the landmark configurations by a Generalized Procrustes Analysis (GPA) of all 113 landmarks, and one in which all landmarks are registered based on a GPA of the 13 landmarks on the body of the first sacral vertebra only (Procrustes fit on a subset of landmarks ^{34,35}). This second geometric morphometric analysis is chosen because the relative size and shape of the sacral wings vary greatly with respect to the sacral body. The datasets originating from the GPA are analysed both in shape space and form space. To assess the morphological differences between the StW 431 and Sts 14 sacra their Procrustes distance are compared to all pairwise differences within our extant species sample.'
2.7	I am a huge advocate of this kind of comparisons in geometric morphometrics but if I am correct and I am not missing anything, the authors did not report the results of the second approach; they only say that "did not obtain relevant differences in the outcomes and thus presented the results for the complete landmark configuration". I personally think it would be convenient to report overall (not necessarily detailed) results of the two analyses (with surface sml vs without surface sml) in supplementary information as Mitteroecker and Bookstein (2008) did in their section "Alternative analyses" of Mitteroecker, P., & Bookstein, F. (2008). The evolutionary role of modularity and integration in the hominoid cranium. Evolution: International Journal of Organic Evolution, 62(4), 943-958.	We hope that the clarification of point 2.6 addresses the current one, too. Note also that the results of the GM analysis after Procrustes fit on a subset of landmarks are discussed in the main article and illustrated in the supplementary information (Fig. S4). Since the outcomes of the analysis without surface landmarks did not differ considerably from the analysis including all landmarks, we do not find this analysis particularly enriching for the supplementary materials. Thus, unless the reviewers and the editor think otherwise, we report the related PCA plot only in this document (Fig. Reply 2.7, below)

2.8	- The authors measured only the first two sacral vertebrae but, if I am correct, they did not explain why. Maybe this is obvious because Sts 14 and StW 431 only preserve that region, but this is not indicated in the manuscript. Relatedly, there is no mention to the different number of sacral vertebrae present in the different species analyzed. If the approach of analyzing the first two sacral vertebrae avoids the problem of lack of homology between the sacra of these taxa, the authors should indicate it.	We specify now in the Methods section that (l. 269-273) “The landmark configuration was conceived to represent the preserved aspects of the StW 431 and the Sts 14 sacrum. It was therefore confined to the first two sacral vertebrae (which are developmentally homologous among all taxa considered in this study)⁵⁶ and consisted of 29 anatomical landmarks, 36 semilandmarks on five curves, and 48 surface semilandmarks (Supplementary Fig. 5).”
2.9	- What do the authors mean by “full GPA”? A GPA with rotation, translation and scaling? A GPA where the centroid size is reduced to cos(ρ) (see Zelditch et al. 2012, Chapter 4)? A GPA superimposing the full configuration (113 landmarks and semilandmarks)?	[Please see our comments 2.6 with respect to the rationale behind the two GM approaches used (GPA vs. Procrustes fit on a landmark subset)]. We changed full GPA to GPA through the manuscript (l. 88, 123, 125, 172, 296, 310, SOM 36, 43). We used the term “GPA” for the standard algorithm that aligns the landmark configurations using all points of the landmark configurations, through translation, rotation and scaling to unit centroid size (“partial Procrustes fitting” sensu Rohlf 1999). The Procrustes fit on a landmark subset aligns the landmark configurations based on a subset of points only, also through translation, rotation and scaling to unit centroid size.
2.10	By the way, what do you mean by “full space” on page 13, line 274?	We meant ‘form space’ (l. 296). We have now corrected our mistake.
2.11	- The authors briefly mention PC3 (e.g. page 7, line 136, page 8 line 138) but if I am correct, this plot is not in the manuscript.	For the analysis of the complete sample, PC3 is shown in the supplementary animation file. A similar animation file (Supplementary data file S2) is now supplied also for the analysis including hominins only.
2.12	- In page 10, lines 183-186 the authors mention perpendicular “vectors” in the PCA analysis but I don’t see any vector in any of the PCA plots. Are these vectors a way to connect means within the PCA? Are statistical vectors?	The vectors we refer to are the vector that would connect StW 431 and Sts 14 as well as the vector from the subadults to the adults in the PCA plot in Fig. 3b. The first one closely aligns with PC 1 and the second with PC 2. As this plot is already quite dense, we did not draw these vectors, but we extended the description in the manuscript, which now sounds (l. 186-190): ‘It is also unlikely that the shape differences between the fossils result from differences in individual age because in our PCA analysis (Fig. 3) the vector between StW 431

		and Sts 14 (which would closely align with PC 1) was almost perpendicular to the vector of average human ontogenetic shape change (close to PC 2).'
2.13	- Page 10, line 185. "In our PCA". Which PCA? Please, indicate Figure.	We added a reference to Fig. 3 (l. 188).
2.14	- Page 11, line 225. "Our worldwide sample 63 modern humans...". Is there a word missing?	It now reads (l. 240-241): 'Our geographically diverse modern human sample comprised 63 individuals...'
2.15	- Page 14, line 285. "using Quadratic Discriminant Analysis". Please, add a reference to Supplementary Note" after this.	Reference to Supplementary Note 1 is added (l. 312)
2.16	"Major" comments - What is the role of the auricular surface in this study? I can see that the morphology is quantified through the landmark configuration (113) but its variation is overlooked in the manuscript. It is surprising as Stw 431 is characterized by its robusticity and body size so assessing this surface would be of interest, basically because the authors reconstructed this part as shown in Figure 1. If the authors consider that this is not relevant for the study, please, explain why (in the text).	We have analysed the sacro-iliac joint and report the outcome below in Fig. Reply 2.16. Please consider also that our goal was to quantify the morphological differences of the sacrum between Sts 14 and StW 431 in a comparative setting to explore whether the magnitude of their shape differences is compatible with within-species variation of modern humans and extant great ape species. Since the morphology of the auricular joint surface does not contribute to this aim, it is not relevant in the context of this study.
2.17	- Did the authors assess the measurement error? Intra- and/or inter-?	Landmarks were collected and carefully double-checked by two experienced observers. Yet, we performed an intra- and inter-observer error assessment and added the following paragraph into the manuscript (l. 301-304): 'An intra- and inter-observer error assessment was performed by C.F. and V.A.K. that confirmed high accuracy and precision of the landmark configuration (highest Procrustes distance between all repeats was lower than 2700 Procrustes Distances out of 2701 possible univocal combinations derived from sample of 74 human sacra).' [(((74x74)-74)/2)=2701] Additionally, analyses were performed independently by two of us.
2.18	- I wonder why it of interest to include Pongo and Gorilla (especially Pongo) in this study. I suspect it is way to standardize the analyses including all great apes, but I wonder why the inclusion of our closest living relatives of the genus Pan is not enough for this kind of comparative studies. Gorilla could be considered by some authors as a scaled version of Pan (similar locomotor mode, etc.) but Pongo is a completely different thing and its inclusion needs to be justified. The bulk of the analyses of	We agree that it is often critical which comparative samples are used or not used. However, again, our goal was only to explore whether the magnitude of the shape difference between Sts 14 and StW 431 is compatible with within-species variation of modern humans and extant great ape species. Therefore, it is important to not only focus on the two Pan species, but also on the degree of variation within Gorilla gorilla and Gorilla beringei as well as within Pongo pygmaeus and Pongo abelii .

	work are PCA and PCA is very sensitive to the specimens included and their variation so a PCA with only Homo and Pan would look different from a PCA including all great apes.	In any case, we do not expect the distribution and relative position of the specimens along the PC axes to change after removing Pongo from the analysis, provided that the rest of the sample and the landmark configurations remain the same. The comparison of the plots (with labels) in Fig. Reply 2.18 below supports this statement.
2.19	To increase the transparency and openness of the reviewing process promoted by the journal, below I sign my report to authors. Lastly, my apologies if any of my comments does not make sense, are redundant or if I missed something after the several readings. I am looking forward to reading a revised version of the manuscript. Nicole Torres-Tamayo	We appreciate this transparency very much and are grateful to Nicole for the useful comments.
3	Reviewer #3 (Remarks to the Author): This paper employed a 3D landmark-based geometric morphometric approach to quantify sacral morphology in extant large hominoids (human, in which to place the differences in sacral morphology observed between Sts 14 and StW 431 to test the hypothesis that these specimens do not represent conspecific individuals. The authors analysis shows that the morphology of these specimens do not exceed the range of variation seen within a single modern specie, albeit they would fall at extreme ends of the range of variation. Therefore, the data do not falsify the hypothesis that these specimens represent a single species. Furthermore, given relatively small sample sizes (Our worldwide sample 63 modern humans including Central Europeans (N=28), Western Africans (N=13), Khoe-Sān (N=7) and Pygmies (N=2), Indians (N=8), and Fuegians (N=5). The great ape sample included G. beringei (N=10), G. gorilla (N=23), P. paniscus (N=8), P. troglodytes (N=22), P. abelii (N=7), and P. pygmaeus (N=24). The differendes between these specimens represents typical morphology associated with some segmental variation within species. As the data and analysis do not falsify the hypothesis of conspecificity, they do not support the	We did not elaborate on the comments of Reviewer 3 since the editor had previously rebutted them on the basis that our results were interpreted with adequate caution within the manuscript's discussion section.

authors conclusions. I therefore recommend this paper be rejected.

Fig. Reply 2.7 - PCA plot for the analysis of the full sample without surface landmarks

Fig. Reply 2.7 - PCA plot for the analysis of the full sample without surface landmarks, after GPA. The outcome is comparable to the one obtained using the full landmark configuration (thus including the surface semilandmarks), shown in the manuscript (Fig. 2).

Fig. Reply 2.16 - PCA analysis in shape space after GPA of the 12-landmark configurations representing the right sacroiliac joint

a)

b)

Fig. Reply 2.16 - PCA analysis in shape space after GPA of the 12-landmark configurations representing the right sacroiliac joint. a) Complete hominoid sample. b) Hominin sample. Based on the shape of the auricular surface, which was only minimally virtually reconstructed in StW 431, the differences between StW 431 and Sts 14 were of minor degree compared to the variability within the reference samples

Fig. Reply 2.18 – Comparison of the PC1-PC2 plot after GPA of the full landmark configuration

Fig. Reply 2.18 – Comparison of the PC1-PC2 plot after GPA of the full landmark configuration. a) with and b) without *Pongo* (*Pongo* males in orange, *Pongo* females in yellow, *Pongo abelii*: PA; *Pongo pygmaeus*: PPYG). The distribution of the rest of the sample does not vary depending on the inclusion of the *Pongo* subsample.

	R. J. Clarke’s comments (via ResearchSquare)	Reply to R. J. Clarke
RJC_1	The title should rather have ‘ Australopithecus africanus ’ in quote marks as you are suggesting heterogeneity, or else simply leave out the africanus species in the title.	New title (l. 1-2 and SOM l. 1-2): ‘Sacrum morphology supports taxonomic heterogeneity of “ Australopithecus africanus ” at Sterkfontein Member 4’ (OR ‘Sacrum morphology supports taxonomic heterogeneity of Australopithecus at Sterkfontein Member 4’ if quotation marks are not allowed)
RJC_2	Towards the end of the paper where you discuss dating and stratigraphy, you need to be very cautious of interpretations of both the dating and of the relationships of the bones in the infills. This is because dates on flowstones have been strongly questioned by Bruxelles et al. (2014, 2019—see our skull paper for these references), because in most cases at Sterkfontein the flowstones are void infills that post-date, perhaps by a million years, some of the infills. The dating of Member 4 is current under new efforts and you should not rely on previous suggestions that Mrs Ples is young, because I know for a fact that the location suggested for Mrs Ples is not correct and that it was not associated with flowstone. I have proof that I have not yet published. The suggested association of Sts 14 with Mrs Ples by Francis is unsubstantiated because there are many other Australopithecus fossils in that same vicinity and there is no reason to assume that the skeleton belongs with Mrs Ples rather than any of the others. Your comments on the distribution of the StW 431 skeletal elements indicating that parts fell into fissures rather than being on a talus slope may not be accurate: However, parts of the StW 431 skeleton were vertically distributed between a depth of 6.5 m below datum (some vertebrae) and a depth of 9.0 m (right pubis fragment) 23. This distribution does not conform to that of a simple talus cone resulting from bones and debris falling into a cave from an above opening as suggested previously 40,41. I believe there is possibly an error in Alun’s catalogue concerning part of the ulna of 431, which I need to check in the vault when I can next go in there. It could be that there was a mistake in the depth assigned to one or more parts of the skeleton. But even so, our experience of the site over many years has shown that there are indeed talus slopes, and there have also been	We addressed these points in the relevant section (Geological age) rephrasing our text as follows (l. 193-208): ‘According to the most recent U-Pd dating of flowstones ^{37,38} the maximum period for the accumulation of the Sterkfontein M4 is between 2.6 and 2.07 Ma, thus Sts 14 and StW 431 could differ up to 540,000 years in chronological age. The exact provenience of Sts 14 is unknown but there are claims that it originated from sediments close to the top of Member 4. This skeleton was said to be found within a single block not far from Sts 5 (‘Mrs. Ples’) ^{39,40} which in turn might have come from the vicinity of the flowstone topping Sterkfontein Member 4 ³⁷ . However, flowstones are often post-depositional infillings of voids within the cave sediments and thus only provide a minimum age for the fossils ⁴¹ . On the other hand, almost all fragments of the StW 431 skeleton were found in two adjacent square yards at a mean depth of 7 m below datum, while many other Member 4 fossils were recovered from deeper deposits. Parts of the StW 431 skeleton were vertically distributed between a depth of 6.5 m below datum (most of the vertebrae and right arm) and a depth of 9.0 m (right pubis fragment) ^{24,42} . This distribution reflects a complex taphonomy of the StW 431 skeleton which might be explained by disturbance of a talus cone by later collapses ^{43,44} . Thus, although the StW 431 remains are challenging to date, there is no indication that they are markedly different in age from Sts 14.’ References within this section: Clarke, R. J. & Kuman, K., 2019 - Robinson, J. T., 1962 were replaced with the more appropriate:

collapse events, so it is possible for parts of one individual to be initially on a talus slope and then to be disturbed by later events and end up at different depths.	43 - Partridge, T. C. & Watt, I. B., 1991 44 – Clarke, R. J., 1994
--	---

Authors' additional changes	
A.1	Abstract: To accommodate the suggestion of Reviewer 1 and keep the abstract within the recommended words limit, we slightly rephrased the abstract to: 'The presence of multiple Australopithecus species at Sterkfontein Member 4, South Africa (2.07–2.61 Ma) is highly contentious, and quantitative assessments of craniodental and postcranial variability remain inconclusive. Using geometric morphometrics, we compared the sacrum of the small-bodied, presumed female subadult Australopithecus africanus skeleton Sts 14 and the large, alleged male adult StW 431 against a geographically diverse sample of modern humans, and two species of Pan , Gorilla , and Pongo . The probabilities of sampling morphologies as distinct as Sts 14 and StW 431 from a single species ranged from 1.3 to 2.5% for the human sample, and from 0.0 to 4.5% for the ape samples, depending on the species and the analysis. Differences in developmental or geologic age or sexual dimorphism could not explain the differences between StW 431 and Sts 14, suggesting that they are unlikely conspecific. This supports earlier claims of taxonomic heterogeneity at Sterkfontein Member 4.'
A.2	Through the manuscript (l. 19, 113, 243), including figures (Legend of Fig. 2, 3a, 4, and Fig. S1, S2), captions of Fig. 4, S4), and Table 1 and 2, we have changed the order in which we present two of the great apes (from ' Gorilla , Pan ' to ' Pan , Gorilla ') to better represent the phylogenetic relationships of great apes to hominins. The new figures and tables are reported at the bottom of this document.
A.3	Lettering in figures and captions has changed from upper to lower-case, as per the Style and formatting guide. (Fig. 1, 3, 4, S5). The new figures are reported at the bottom of this document.
A.4	To comply with the Style and formatting guide, we added a conclusive sentence at the end of the Introduction (l. 83-86): 'This investigation revealed that the morphological differences between Sts 14 and StW 431 could not be fully explained by developmental or geologic age or sexual variation suggesting that taxonomic heterogeneity might be considered to interpret the morphological variability within Sterkfontein Member 4 Australopithecus .'
A.5	We deleted the redundant information in l. 174-177: 'Our study uses a comprehensive geometric morphometric approach to quantitatively assess the morphological variation within the Sterkfontein Member 4 Australopithecus sacrum sample against the background of a large, worldwide sample of modern humans and six great ape species' and l. 180-181: 'by the shape differences between'
A.6	We rephrased part of our Discussion in l. 220-229: 'Since Sts 14 and StW 431 preserve only the first two and a half sacral vertebrae, it is impossible to completely rule out transitional vertebrae in either of these specimens, hence this aspect should be investigated further. In fact, Sts 14 shows a segmentation anomaly at the thoracolumbar transition. Border shifts of the thoracolumbar junction are frequent in hominin fossils and are often associated with border shifts at the lumbosacral junction 50. The morphological differences between the partial sacra of StW 431 and Sts 14 are remarkable when compared to the range of variation in modern humans and great apes. Although these morphological discrepancies can be partly explained by individual variation, further exploration of the impact of sexual dimorphism, allometry, different individual and geological age, and, eventually, taxonomic heterogeneity are necessary.'

A.7

According to the Style and formatting guide, 'Main' is now 'Introduction', and the Methods' section 'Geometric morphometric analysis' was changed into 'Statistics and reproducibility'

FIGURES:

In figures' legend of Fig. 2, 3a, 4, and Fig. S1, S2, captions of Fig. 4, S4), we have changed the order in which we present two of the great apes (from 'Gorilla, Pan' to 'Pan, Gorilla') to better represent the phylogenetic relationships of great apes to hominins. Lettering in figures and captions has changed from upper to lower-case, as per the Style and formatting guide. (Fig. 1, 3, 4, S5). The 13 landmarks used for the Procrustes fit superimposition are now shown in Supplementary Figure 5 c) and d). Figures are reported below:

Figure 1: The sacrum of Sts 14 and StW 431, both traditionally attributed to *A.*

africanus. **a)** Photograph of the Sts 14 sacrum, superior view and **b)** anterior view. **c)** 3D surface model of the reconstructed Sts 14 sacrum produced by mirroring the left side with respect to the midsagittal plane (red line), thereby removing the plaster reconstruction of ²¹ (shown in transparent). **d)** Photograph of the StW 431 sacrum, superior view and **e)** anterior view. **f)** Reconstructed 3D surface model of StW 431 (in transparent) obtained by mirroring the left side of the sacrum at the mid-sagittal plane (red line). **g)** Photograph and **h)** 3D surface model of the left ilium fragment of StW 431; the sacroiliac joint surface is coloured in red. **i)** The most inferior portion of the sacral auricular surface (arrow) was restored using the auricular surface of the ilium.

Figure 2: PCA plot of the shape coordinates of the first two sacral vertebrae and auricular surface after a ~~full~~ Generalized Procrustes Analysis (GPA). Sts 14 and StW 431 (both attributed to *Australopithecus africanus*) plot at opposite sides of the modern human distribution, while the two reconstructions of the A.L. 288-1 (*A. afarensis*) sacrum are close to Sts 14. The great apes are separated from the hominins along PC1. PC1 is driven by the overall height-to-width ratio of the sacrum, and PC2 represents the orientation and relative antero-posterior width of the sacral alae.

Figure 3: PCA plot of the Procrustes shape coordinates for the upper portion of the sacrum (full-GPA) in great apes and modern humans. a) PCA plot of great apes, labelled by species and sex. Within each genus, the species and sexes largely overlap, except for *G. beringei*. **b)** PCA plot for the upper portion of the sacrum (full-GPA) in modern humans and *Australopithecus*, labelled by sex, age, and ethnicity. Along PC1, the sacral portion of the linea terminalis varies from more horizontal, as in Sts 14 and A.L. 288-1, to more caudally

oriented, as in StW 431. A.L. 288-1 differs from StW 431 and Sts 14 along PC2 for its posterior orientation of the alae. The various modern human populations do not separate, but the subadult females tend to differ from the adults along PC2, while StW 431 and Sts 14 differ along PC1.

Figure 4: Column distribution of the male-female pairwise Procrustes distances after Procrustes fit based on 13 landmarks on the first sacral vertebra. The red lines indicate the Procrustes distance between Sts 14 and StW 431. **a)** In modern humans only 35 out of 1353 Procrustes distances (2.6%) exceeded that between Sts 14 and StW 431. **b)** In *Pan troglodytes* 0% (0 out of 120) distances exceeded that between the fossils, in *Gorilla gorilla* 4.5% (6 out of 132), and in *Pongo pygmaeus* 3.9% (5 out of 128).

Supplementary Figure 1: Boxplots of the ln centroid sizes. Sts 14 and A.L. 288-1 are smaller than any recent modern humans considered, while StW 431 is at the lower end of the modern human distribution. The ln centroid sizes of the *Australopithecus* sacra are in the range of *Pan* and *Pongo*, and of the smallest *Gorilla* specimens. Sts 14 is smaller than any modern human specimens including the subadults, and falls in the range of variation of *Pan* and *Pongo*, while StW 431 is at the lower end of the distribution of *Homo*. The regression analysis shows that the influence of size on shape is appreciable for this sample (12.3% of the total variance), while it reduces to 3.3% for the hominin subsample.

Supplementary Figure 2: PCA plot in form space after full-GPA for the complete sample including both modern humans and great apes. In this plot, modern humans separate completely from great apes. In this analysis, the male gorillas and orangutans tend to separate from the rest of the great apes for their large size. There is still a complete overlap of male and female modern humans and *Pan*, while a dimorphic trend is observed both in *Pongo* and especially *Gorilla*. The *Australopithecus* individuals are distinct from the rest of the sample.

Supplementary Figure 3: PCA plot in shape space after full-GPA for the modern human adults, Sts 14 and StW 431. In the PC1-PC2 plot, male and female modern humans form two overlapping groups. Sts 14 and StW 431 diverge largely along PC1, which reflects the different cranio-caudal orientation of the wings. When the Procrustes distances are calculated based on this PC, only 1.0% of the pairwise comparisons between all modern humans and 1.1% between the male-female pairs are higher than the distances between Sts 14 and StW 431. The sexual-related variation, corresponding to the relative width of the wings with respect to the sacral body, occurs in PC2, along which Sts 14 and StW 431 also diverge.

Supplementary Figure 4: PCA plot of the shape coordinates after Procrustes superimposition based on a subset of 13 landmarks on the body of the first sacral vertebra. This analysis emphasizes the relative dimensions of the wings in relation to the sacral body. Sts 14 plots at an extreme of the modern human distribution and StW 431 plots on the other side, between modern humans and great apes. These outcomes highlight the large differences between Sts 14 and StW 431 in terms of width-to-height proportions. Modern humans are distinct from the great apes in the PC1-PC2 plot, while *Pan*, *Gorilla*, and *Pongo* overlap extensively.

Supplementary Figure 5: The landmark configuration used. **a)** Anterior and **b)** posterior views of a modern human sacrum. Landmarks: red points; curve semilandmarks: blue points; surface semilandmarks: green points. **c)** anterior and **d)** posterior views showing the landmark subset used for the Procrustes fit (orange).

TABLES:

In Table 1 and 2, we have changed the order in which we present two of the great apes (from '*Gorilla, Pan*' to '*Pan, Gorilla*') to better represent the phylogenetic relationships of great apes to hominins.

Table 1: Percentages of the pairwise Procrustes distances that exceeded the Procrustes distance between StW 431 and Sts 14. The computations were performed for male-female pairs only as well as for all pairwise comparisons. m/f = male-female; S1 = first sacral vertebra

Groups	GPA, shape space		GPA, form space		Procrustes fit on S1 body, shape space	
	m/f pairs	all pairs	m/f pairs	all pairs	m/f pairs	all pairs
Modern human adults (N=63)	1.5	1.3	3.8	3.7	2.6	2.1
Modern human adults and subadults (N=74)	2.5	2.4	3.4	3.3	2.6	2.3
Pan (N=30)	24.1	21.8	6.0	6.9	0.9	1.1
Gorilla (N=33)	18.8	18.8	54.2	39.4	5.4	4.2
Pongo (N=31)	26.7	28.8	39.5	28.2	2.4	4.5
P. troglodytes (N=22)	24.2	22.1	0.8	0.4	0.0	0.0
G. gorilla (N=23)	15.2	15.8	47.7	32.0	4.5	3.6
P. pygmaeus (N=24)	30.5	31.2	43.8	31.5	3.9	6.2

Table 2: List of fossil sacra and modern comparative material

	Individuals / taxa	Females	Males	Collections
Australopithecus	Sts 14q	1		a
	StW 431h		1	b
	A.L. 288-1an	1		c
Modern humans	adults	28	35	d, e, f, g, h, i, j
	subadults	5	6	k
Great apes	Pan paniscus	6	2	l
	Pan troglodytes	12	10	d, h, m, n, o, p, q
	Gorilla beringei	2	8	d, h, m, o
	Gorilla gorilla	11	12	d, h, m, n, o
	Pongo abelii	5	2	d, h
	Pongo pygmaeus	16	8	d, h, l, n, q
Total modern comparative sample (N=168)		85	83	

a) Ditsong National Museum of Natural History, Pretoria; b) Evolutionary Studies Institute, University of the Witwatersrand, Johannesburg, South Africa; c) National Museum of Ethiopia, Addis Ababa, Ethiopia; d) Anthropological Institute and Museum, University of Zurich, Switzerland; e) Department of Anthropology and Narrenturm, Natural History Museum of Vienna, Austria; f) Department of Evolutionary Anthropology, University of Vienna, Austria; g) Institute of Evolutionary Medicine, University of Zurich, Switzerland; h) Laboratory of Prehistoric Archaeology and Anthropology, University of Geneva, Switzerland; i) Museum of Natural History, University of Florence, Italy; j) Smithsonian National Museum of Natural History, Washington, USA; k) Hospital Timone, Marseille, France; l) **Royal Museum for Central Africa, Tervuren, Belgium**; m) Zoological Museum, University of Zurich, Switzerland; n) Digital Morphology Museum, KUPRI, Kyoto University, Japan; o) **Department of Zoology, Natural History Museum of Vienna, Austria**; p) Museum of Primatology, University of California, San Diego; q) Natural History Museum Basel, Switzerland.

REVIEWERS' COMMENTS:

Reviewer #2 (Remarks to the Author):

Thank you very much for inviting me to read a revised version of the manuscript. I consider that the authors addressed my comments and suggestions so, in my opinion, the manuscript is ready to be accepted.

My best wishes for the New Year!

Nicole Torres-Tamayo